# Ecological Niche and Interspecific Association of the Main Fishes in the Coastal Waters of Hainan Island, China

**Zhengli Luo [1,2], Yanbo Zhou [1,3,*], Ning Liu [4], Liangming Wang [1,5], Yan Liu [1,5], Binbin Shan [1,5], Manting Liu [1], Cheng Chen [1], Changping Yang [1,5] and Dianrong Sun [1,5,*]**

1   South China Sea Fisheries Research Institute, Chinese Academy of Fisheries Sciences, Guangzhou 510300, China; luohua0303@163.com (Z.L.); wangliangming@scsfri.ac.cn (L.W.); liuyan@scsfri.ac.cn (Y.L.); shanbinbin@scsfri.ac.cn (B.S.); liumanting@scsfri.ac.cn (M.L.); chencheng@scsfri.ac.cn (C.C.); yangchangping@scsfri.ac.cn (C.Y.)
2   School of Fisheries, Zhejiang Ocean University, Zhoushan 316022, China
3   Key Laboratory for Sustainable Utilization of Open-Sea Fishery, Ministry of Agriculture and Rural Affairs, Guangzhou 510300, China
4   Research Center of Fisheries Resource and Environment, Chinese Academy of Fishery Sciences, Beijing 100141, China; annening@163.com
5   Key Laboratory of Marine Ranching, Ministry of Agriculture and Rural Affairs, Guangzhou 510300, China
*   Correspondence: zhouyanbo@scsfri.ac.cn (Y.Z.); drsun73@scsfri.ac.cn (D.S.)

**Abstract:** This study explored the ecological niche and interspecific relationships among the main fish species in the coastal waters of Hainan Island based on data from fishery stock surveys conducted in the spring and autumn of 2022. The methods of chi-square test, percentage of co-occurrence (*PC*), association coefficient (*AC*), cluster analysis, and redundancy analysis were used to analyze the interspecific associations and influence of environmental factors on the ecological niches of fish communities. According to the cluster analysis of niche breadth, 20 main fishes could be divided into broad, medium, and narrow niche species. There were four and eight broad niche species in spring and autumn, respectively. The ranges of niche overlap values were 0.001–0.91 in spring and 0–0.87 in autumn, indicating that the species differed and were similar in their ability to utilize resources, survive in habitats, and prey. According to the variance ratio and statistic value *W*, the main fishes in spring showed a significant positive association, whereas those in autumn showed a positive association, but not at a significant level, indicating that the main fishes in spring were more closely associated with each other. There were 56 species pairs in the 2 seasons that were significantly associated ($\chi^2 \geq 3.841$). *AC* and *PC* tests revealed that the interspecific association was strong and tended to be positive. According to the redundancy analysis, environmental factors such as surface temperature, water depth, and pH significantly affected the main fishes in spring, while environmental factors such as dissolved oxygen, bottom temperature, surface salinity, and pH significantly affected those in autumn.

**Keywords:** fish community; Hainan Island; ecological niche; interspecific association; redundancy analysis; similarity analysis

**Key Contribution:** This study represents the inaugural investigation and assessment of the ecological niche and interspecific connectivity among major fish species in the nearshore waters surrounding Hainan Island. It establishes the groundwork for a thorough examination of fish community patterns and interactions within varying populations inhabiting the nearshore waters of Hainan Island. Our study unveiled dissimilarities in the utilization of resources, habitat requirements, and predation needs among different fish species. The study suggests a higher degree of connectivity among major fish species during the spring season. Environmental factors such as temperature, salinity, and pH had a significant impact on major fish species. These findings address the knowledge gap in the understanding of fish populations in the nearshore waters of Hainan Island while providing a basic dataset and scientific framework to study the adaptive mechanisms of fish species in response to environmental changes.

## 1. Introduction

Niche theory has been widely used in the fields of interspecific association, community structure, biodiversity, and population evolution as one of the most important general theories of ecology [1,2]. The interspecific association refers to the correlation between the spatial distributions of several species within a habitat as a result of their mutual influence, encompassing not only the number and structural characteristics of the community but also the composition and evolution of the community [3,4].

The concept of ecological niche was first defined by Grinnell [5] in his study of the ecological niche of the California thrasher. Afterward, scholars such as Elton [6], Hutchinson [7], and MacArthur [8] enriched the niche theory through the proposal of nutritional niche, n-dimensional hyper-volume, resource utilization function, and extended niche theories. Ecological niche plays a significant role in the study of community structure and function, interspecific association, biodiversity, the succession of communities over time, and the evolution of populations. Consequently, it is widely used in research and has produced many positive results. The initial studies on ecological niches and interspecific associations in China focused primarily on terrestrial organisms and plant groups [9–11], while studies on aquatic ecosystems emphasized intertidal benthic communities [12,13]. Currently, the focus of research has been on fish [14,15], marine nekton [16], and other marine organisms, covering the Bohai Sea, Yellow Sea, and the central and southern East China Sea.

Previous research has demonstrated that long-term intensive fishing [17], coastal engineering activities [18], and degradation of water quality [19] have resulted in substantial alterations to fish community structure and biodiversity in coastal waters. Extensive research has been conducted on the composition of the fish communities and the biodiversity of fish populations to monitor ecological conditions and assess resource availability [20]. However, the fisheries resources in the coastal waters of Hainan Island have only been studied by Sun et al. [21] and Zhang et al. [22]. Furthermore, no information is available regarding the ecological niche and interspecific association of the main fish species found in this region.

This paper examines the niche width, niche overlap, and interspecific connectivity measures of major fish species using data from surveys conducted in the nearshore waters of Hainan Island during spring and autumn 2022. Additionally, a redundancy analysis was conducted to assess the impact of environmental factors on major fish species. The study aimed to achieve the following objectives: (1) understanding the extent of resource utilization and the size of ecological niches occupied by major fish species within the ecosystem; (2) exploring major fish species patterns of potential competitive relationships and resource allocation through analysis of ecological niche overlap; (3) revealing patterns and strengths of species interactions within fish communities and ecosystems; and (4) understanding the influence of environmental factors on fish community structure and species interactions. The research objectives seek to enhance our understanding of the ecological niche characteristics of major fish species, interspecies interactions, and the influence of environmental factors, with a potentially positive impact on the conservation and management of aquatic resources and the maintenance of ecosystem stability.

## 2. Methods and Materials

### 2.1. Study Area

As a large offshore island of China (108°37′00″–111°03′00″ E and 18°10′00″–20°10′00″ N), Hainan Island is located in tropical and subtropical waters, facing Leizhou Peninsula across the Qiongzhou Strait to the north, and Qinzhou and Vietnam to the west across the Beibu Gulf. Because of its exceptional geographical location and complex hydrological conditions, its offshore fishing grounds are among the most important in the South China Sea. A rich source of marine biological resources [21], the area is also an important spawning, feeding, and rearing area for marine fishes.

## 2.2. Material Sources

In this study, data were obtained from two bottom-trawl surveys conducted in May (spring) and September (autumn) 2022 in the coastal waters of Hainan Island at a depth of 15–160 m (Figure 1). The fishery resource data in this study are quoted from our recently published research paper [23]. A total of 50 stations were surveyed, and all of the trawlings were conducted during the day at an average trawling speed of 3 knots for approximately 1 h. The survey vessel used in this study was the bottom trawler "Guibeiyu 69068" with a main engine power of 436 Kw, a length of 53.8 m, a molded breadth of 8.2 m, a molded depth of 4.6 m, and a gross tonnage of 590 t. The survey nets were 404-mesh bottom trawl nets with a net opening breadth of 37.7 m, a net opening mesh size of 20 cm, and a mesh bag mesh size of 40 mm.

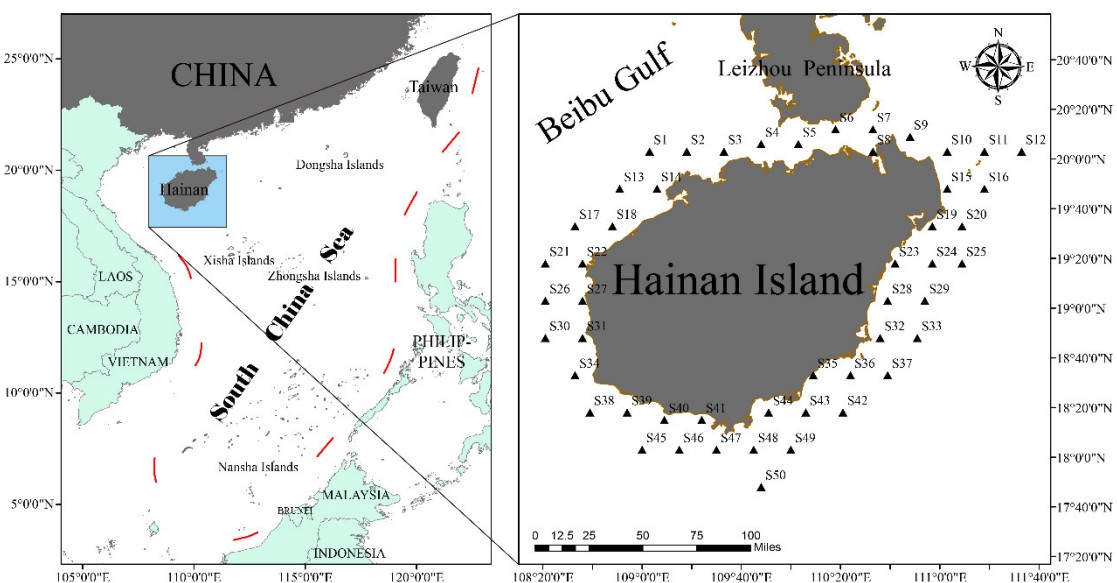

**Figure 1.** Survey stations for fish resources in the inshore waters of Hainan Island.

The on-station investigation followed the "Technical Regulations for Marine Biological Ecological Survey", and the collection and analysis of catches followed the "Specifications for the Investigation of Marine Fishery Resources". The survey area encompassed coordinates ranging from 108°21′00″ to 111°33′00″ E and 17°47′00″ to 20°12′00″ N. Fish classification and identification, counting, and body weight determination (accurate to 0.1 g) were carried out on the catches. Fish taxonomic identification referred to Search FishBase [24], "Key to Marine and Estuarial Fishes of China" [25], "Marine Fishes of China" [26], and "Fishes of Taiwan" [27], and the fish were identified to the lowest possible taxonomic level. YSI 5908 MEMBRANE KIT-12.5 MILPE was used in the trawl survey to measure water temperature, salinity, pH, dissolved oxygen, and other environmental parameters at each station synchronously. The concentration data of Chl-a came from NASA Ocean Color [28], and the water depth was surveyed by the depth gauge that came with the hull. Finally, the resulting data were standardized.

## 2.3. Dominance Analysis

### 2.3.1. The Index of Relative Importance (IRI)

The *IRI* index [20,29,30] was used to measure the ecological dominance of fish communities in each season.

$$IRI = (N + W) \times F \times 10^4 \tag{1}$$

where $N$ (%) is the percentage of individuals of a certain fish species accounting for the individuals of total catch; $W$ (%) is the percentage of the wet weight of a certain fish species accounting for the wet weight of total catch; and $F$ (%) is the percentage of stations where the fish species appeared in the total number of stations. Dominant species: $IRI \geq 1000$, important species: $100 \leq IRI < 1000$, common species: $10 \leq IRI < 100$, and rare species: $IRI < 10$. Because the dominant species and important species that appeared in the two surveys had a relatively large number and a mass proportion, both were defined as main fishes in this study.

### 2.3.2. Ecological Niche

The term niche overlap refers to the similarity or duplication of resource utilization between two or more species [31]. A niche overlap is also an important aspect of biodiversity, which is a reflection of the interaction and competition between different species and has a significant impact on the stability and function of an ecosystem.

Niche breadth was determined using the Shannon index [32].

$$B_i = - \sum_{j=1}^{R} (P_{ij} \, In P_{ij}) \tag{2}$$

The niche overlap index was determined by the Piankas index [33].

$$Q_{ik} = \sum_{j=1}^{R} (P_{ij} \, P_{kj}) / \sqrt{\sum_{j=1}^{R} P_{ij}^2 \sum_{j=1}^{R} P_{kj}^2} \tag{3}$$

where $P_{ij}$ and $P_{kj}$ are the proportions of species $i$ and $k$ in the total number of fish at station $j$; $R$ is the total number of stations; $B_i$ is the niche breadth index, which ranges between 0 and $R$, and the larger the value is, the broader the niche of the species is; and $Q_{ik}$ is the niche overlap index, which ranges between 0 and 1.

### 2.3.3. Interspecific Association

The variance ratio proposed by Schluter [34] is an index for determining the overall association between species. The statistical value $W$ was used to test the significance of the association as follows.

$$\delta_T^2 = \sum_{i=1}^{s} P_i \, (1 - P_i)^2 \tag{4}$$

$$S_T^2 = \frac{1}{n} \sum_{j=1}^{n} (T_j - t)^2 \tag{5}$$

$$VR = S_T^2 / \delta_T^2 \tag{6}$$

$$W = VR \times n \tag{7}$$

where $\delta_T^2$ represents the overall variance of total station number; $S_T^2$ represents the overall variance of total species number; $P_i$ is the occurrence frequency of the *ith* species, $P_i = n_i/n$; $n$ represents the number of stations; $n_i$ represents the number of stations where species $i$ occurs; $T_j$ stands for the species number of main fishes at station $j$; s represents the total species number of main fishes; and $t$ denotes the average number of species at

station *j*. *VR* = 1 indicates no association between species; *VR* > 1 indicates a positive correlation between species; *VR* < 1 indicates a negative correlation between species. The statistic value *W* was used to test the significance level of *VR* value deviating from 1; if the value *W* falls into the confidence interval $\chi^2_{0.95}(n) < W < \chi^2_{0.05}(n)$ of the chi-square test, there is no association between species with a probability of 90% confidence.

The chi-square test was based on a 2 × 2 contingency table, tested with the Yates continuous correction method [35,36].

$$\chi^2 = \frac{n(|ad - bd| - 0.5n)^2}{(a + b)(b + d)(a + c)(c + d)} \tag{8}$$

where *n* is the total number of stations; *a* is the number of stations where both species appear; *b* and *c* are the numbers of stations where only one of the species appears; and *d* is the number of stations where neither species appears.

As for the association coefficient (*AC*) [37],

$$\text{If } ad \geq bc, \text{ then } AC = (ad - bc)/(a + b)(b + d) \tag{9}$$

$$\text{If } bc > ad, d \geq a, \text{ then } AC = (ad - bc)/(a + b)(a + c) \tag{10}$$

$$\text{If } bc > ad, a > d, \text{ then } C = (ad - bc)/(b + d)(c + d) \tag{11}$$

In Equations (9)–(11), the value of *AC* ranges from −1 to 1; the closer the *AC* value is to 1, the stronger the positive association between the species pairs; the closer the *AC* value to −1, the stronger negative association between the species pairs; when *AC* is 0, the species are independent of each other.

The percentage of co-occurrence (*PC*) [38] was determined as follows.

$$PC = a/(a + b + c) \tag{12}$$

In Equation (12), the *PC* value ranges from 0 to 1. The closer the *PC* value is to 1, the stronger the positive association between species is. When the species pairs are independent of each other, the *PC* value is 0. The meanings of a, b, c, and *d* refer to Formula (8).

### 2.3.4. Redundancy Analysis

Detrended correspondence analysis (DCA) was first performed on the species data, and the best sorting method was determined according to the gradient length (LGA) of each axis. When LGA < 3, redundancy analysis (RDA) was used; when LGA > 4, canonical correspondence analysis (CCA) was used; and when 3 < LGA < 4, both were acceptable [39]. According to the results of DCA, this study employed RDA to analyze the environmental factors and the spatial distribution of main fishes to reveal their niche differentiation in this sea area. RDA allows comprehensive analysis of multiple environmental factors, which can directly reflect the correlation between species communities and various environmental factors ($\alpha$ = 0.05).

### 2.3.5. Data Processing

Arcgis 10.8 was used to visualize the survey stations and mark the relevant geographic coordinates. The *IRI* of species was calculated by Excel 2016 and the ecological niche was calculated in the "spaa" package of the *R* language. Primer 5.0 was used for cluster analysis; RDA was performed on environmental factors by Canoco 5.

## 3. Results

### 3.1. Composition of Catches

A total of 363 species of fish, belonging to 24 orders, 114 families, and 226 genera (Table S1), were caught in the 2 bottom-trawl surveys in the coastal waters of Hainan Island. According to the relative importance index (*IRI*), there are 20 main fishes, including 9 species in spring and 17 species in autumn. The dominant species in spring (*IRI* ≥ 1000) were *Acropoma japonicum* and *Decapterus maruadsi*. There were four important species (100 ≤ *IRI* < 1000) in the two seasons, namely, *Upeneus japonicus*, *Saurida tumbil*, *Champsodon atridorsalis* and *Saurida undosquamis* (Table 1).

**Table 1.** Relative importance index (*IRI*) of major fishes and their ecological niche width (*Bᵢ*). * Represents species shared in spring and autumn.

| Species | Dominant Species | Spring | | Autumn | |
|---|---|---|---|---|---|
| | | *IRI* | *Bᵢ* | *IRI* | *Bᵢ* |
| S1 * | *Acropoma japonicum* | 3013.82 | 1.494 | 332.49 | 2.263 |
| S2 * | *Decapterus maruadsi* | 1475.74 | 0.621 | 200.44 | 2.773 |
| S3 | *Navodon xanthopterus* | 548.26 | 2.992 | | |
| S4 | *Trachurus japonicus* | 386.92 | 1.374 | | |
| S5 * | *Upeneus japonicus* | 385.01 | 2.027 | 331.61 | 2.331 |
| S6 | *Psenopsis anomala* | 321.09 | 2.239 | | |
| S7 * | *Saurida tumbil* | 217.72 | 3.039 | 646.08 | 3.293 |
| S8* | *Champsodon atridorsalis* | 194.26 | 2.994 | 528.74 | 2.863 |
| S9 * | *Saurida undosquamis* | 147.97 | 2.962 | 380.45 | 3.216 |
| S10 | *Leiognathus bindus* | | | 730.84 | 2.824 |
| S11 | *Leiognathus berbis* | | | 451.21 | 2.571 |
| S12 | *Johnius belengeri* | | | 392.12 | 2.071 |
| S13 | *Pennahia macrocephalus* | | | 372.31 | 1.978 |
| S14 | *Brachypleura novaezeelandiae* | | | 256.09 | 2.715 |
| S15 | *Pennahia anea* | | | 240.42 | 2.165 |
| S16 | *Therapon thraps* | | | 178.53 | 2.259 |
| S17 | *Rogadius asper* | | | 145.78 | 0.297 |
| S18 | *Upeneus sulphureus* | | | 118.03 | 2.592 |
| S19 | *Ilisha melastoma* | | | 113.01 | 1.909 |
| S20 | *Parargyrops edita* | | | 108.69 | 2.143 |

### 3.2. Main Species and Niche Breadth

The niche breadth of the main fishes in the coastal waters of Hainan Island ranged between 0.297 and 3.293, with significant differences and a staged distribution from low to high. In spring, *S. tumbil* had the greatest niche breadth (3.039), followed by *C. atridorsalis* (2.994), and *D. maruadsi* had the lowest niche breadth (0.621); in autumn, *S. tumbil* had the highest niche breadth (3.293), *followed by S. undosquamis* (3.216), and *Rogadius asper* had the lowest niche breadth (0.297), as shown in Table 1.

As niche breadth has not been standardized in a unified and consistent manner, in this study niche breadth was classified in accordance with the study by Herawati [40] and the specific niche breadth of the main fish species in the study area, namely, broad niche species: *Bᵢ* ≥ 2.571, medium niche species: 2.027 ≤ *Bᵢ* < 2.571, and narrow niche species: 0.297 < *Bᵢ* < 2.027. In this study, a significant staged distribution was observed in the niche breadth of the main fishes. In spring, there were four broad niche species (section A): *S. tumbil*, *Thamnaconus hypargyreus*, *C. atridorsalis* and *S. undosquamis*, two medium niche species (section B): *U. japonicus* and *Psenopsis anomala*, and three narrow niche species (section C): *A. japonicum*, *D. maruadsi* and *Trachurus japonicus* (Figure 2a). In autumn, there were eight broad niche species (section a), including *Leiognathus bindus*, *Leiognathus berbis*

and *C. atridorsalis*, six medium niche species (section b), including *A. japonicum*, *Johnius belangerii* and *Pennahia anea*, and three narrow niche species (section c): *Pennahia macrocephalus*, *R. asper* and *Ilisha melastoma* (Figure 2b). The results of cluster analysis showed that the niche breadth of the main fishes in the surveyed sea area in spring could be divided into three groups (Figure 3a): *U. japonicus* and *P. anomala* (Group 1); *S. tumbil*, *N. xanthopterus*, *C. atridorsalis*, etc., (Group 2); and *A. japonicum*, *D. maruadsi* and *T. japonicus* (Group 3). The niche breadth of the main fishes in autumn could also be divided into three groups (Figure 3b): *L. bindus*, *L. berbis*, *C. atridorsalis*, etc., (Group 1); *A. japonicum*, *J. belangerii*, *P. anea*, etc., (Group 2); *P. macrocephalus*, *R. asper*, and *I. melastoma* (Group 3).

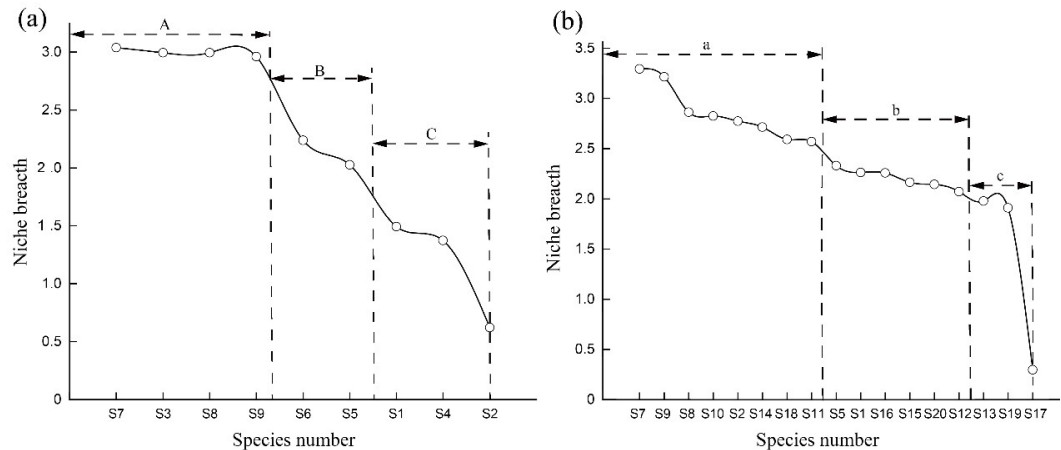

**Figure 2.** Variation in width values of major fish ecological niches in spring (**a**) and autumn (**b**). Species number refers to Table 1. The same below.

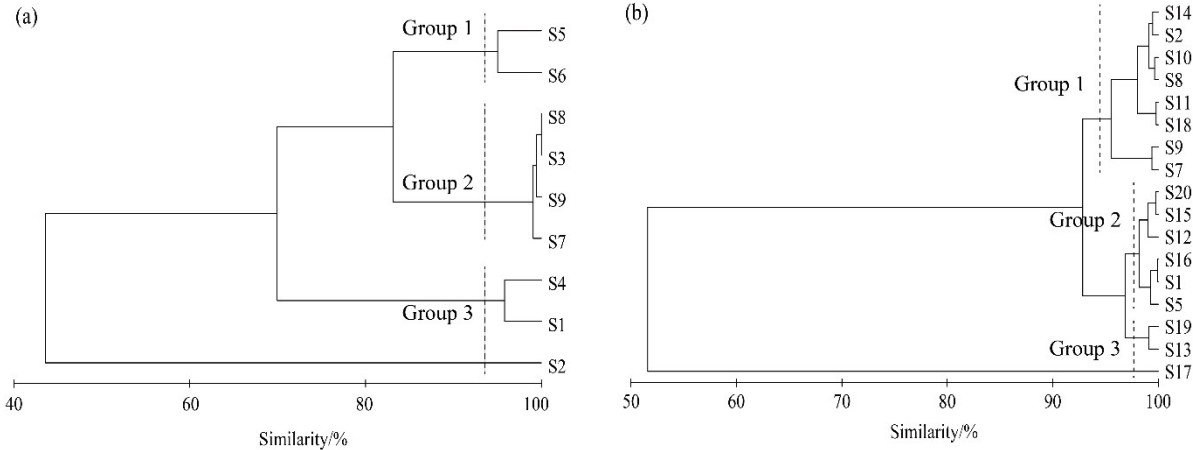

**Figure 3.** Cluster analysis of the width of the main fish ecological niches in spring (**a**) and autumn (**b**).

### 3.3. Niche Overlap

As shown in the study, niche overlap between the main fish species in the coastal waters of Hainan Island was uneven in spring and autumn. In spring, the niche overlap value ranged from 0.001 to 0.91 (Table 2), among which three species pairs had an overlap value less than 0.001, accounting for 8.33% of the total number of pairs, and the species pair *S. tumbil* and *S. undosquamis* (S7–S9) had the greatest niche overlap (0.91); in autumn, the niche overlap value was between 0 and 0.87 (Table 3), and four species pairs had an overlap value less than 0.001, accounting for 2.94% of the total number of pairs, and the

species pair *S. tumbil* and *S. undosquamis* (S7–S9) had the greatest niche overlap (0.87), indicating that the species pair had a high degree of similarity in resource utilization. Wathne et al. [41] found that interspecific niche overlap is significant when the overlap value is ≥0.6. Of the 36 species pairs in spring, there were 3 species pairs with an overlap value of ≥0.8, 20 pairs with an overlap value of <0.2, and 3 pairs with a significant overlap value, which represents 8.33% of the total number of species pairs. A total of 1 species pair out of 136 species pairs in autumn had an overlap value of ≥0.8, 85 species pairs had an overlap value of <0.2, and 8 species pairs had a significant overlap value, which accounted for 5.88% of the total number of species pairs. The niche overlap values in both spring and autumn were low, and in general, niche overlap values in spring were slightly higher than those in autumn.

**Table 2.** Overlap values of major fish ecological niches in spring.

| Species | S1 | S2 | S3 | S4 | S5 | S6 | S7 | S8 |
|---|---|---|---|---|---|---|---|---|
| S2 | <0.001 | | | | | | | |
| S3 | 0.096 | 0.027 | | | | | | |
| S4 | 0.006 | 0.01 | 0.36 | | | | | |
| S5 | <0.001 | <0.001 | 0.17 | 0.01 | | | | |
| S6 | 0.035 | 0.41 | 0.30 | 0.86 | 0.004 | | | |
| S7 | 0.01 | 0.077 | 0.31 | 0.22 | 0.21 | 0.33 | | |
| S8 | 0.01 | 0.027 | 0.83 | 0.36 | 0.17 | 0.30 | 0.31 | |
| S9 | 0.038 | 0.047 | 0.32 | 0.067 | 0.27 | 0.17 | 0.91 | 0.32 |

**Table 3.** Overlap values of major fish ecological niches in autumn.

| Species | S10 | S7 | S8 | S11 | S12 | S9 | S13 | S1 | S5 | S14 | S15 | S2 | S16 | S17 | S18 | S19 |
|---|---|---|---|---|---|---|---|---|---|---|---|---|---|---|---|---|
| S7 | 0.13 | | | | | | | | | | | | | | | |
| S8 | 0.20 | 0.60 | | | | | | | | | | | | | | |
| S11 | 0.28 | 0.16 | 0.21 | | | | | | | | | | | | | |
| S12 | 0.11 | 0.002 | 0.004 | 0.17 | | | | | | | | | | | | |
| S9 | 0.13 | 0.87 | 0.61 | 0.13 | 0.003 | | | | | | | | | | | |
| S13 | 0.29 | 0.05 | 0.08 | 0.51 | 0.21 | 0.02 | | | | | | | | | | |
| S1 | 0.42 | 0.14 | 0.17 | 0.44 | 0.05 | 0.11 | 0.71 | | | | | | | | | |
| S5 | 0.02 | 0.34 | 0.24 | 0.10 | 0.001 | 0.49 | 0.01 | 0.07 | | | | | | | | |
| S14 | 0.25 | 0.47 | 0.28 | 0.42 | 0.01 | 0.36 | 0.63 | 0.64 | 0.04 | | | | | | | |
| S15 | 0.30 | 0.01 | 0.004 | 0.04 | 0.58 | 0.01 | 0.10 | 0.13 | 0.002 | 0.01 | | | | | | |
| S2 | 0.36 | 0.17 | 0.23 | 0.04 | 0.22 | 0.19 | 0.05 | 0.51 | 0.09 | 0.07 | 0.24 | | | | | |
| S16 | 0.39 | 0.02 | 0.01 | 0.08 | 0.32 | 0.01 | 0.32 | 0.03 | 0.02 | 0.005 | 0.20 | 0.13 | | | | |
| S17 | 0.14 | 0.05 | 0.08 | 0.40 | 0.01 | 0.03 | 0.63 | 0.55 | 0.003 | 0.69 | 0 | 0.001 | <0.001 | | | |
| S18 | 0.61 | 0.07 | 0.09 | 0.41 | 0.18 | 0.05 | 0.26 | 0.21 | 0.03 | 0.09 | 0.13 | 0.22 | 0.59 | 0.07 | | |
| S19 | 0.29 | 0.01 | 0.04 | 0.09 | 0.35 | 0.002 | 0.05 | 0.002 | <0.001 | 0.01 | 0.32 | 0.28 | 0.16 | <0.001 | 0.18 | |
| S20 | 0.21 | 0.11 | 0.01 | 0.39 | 0.04 | 0.08 | 0.03 | 0.20 | 0.08 | 0.07 | 0.08 | 0.11 | 0.05 | <0.001 | 0.21 | 0.17 |

### 3.4. Overall Association Analysis

The variance ratio (*VR*) was used to test the overall association of the main fishes in the coastal waters of Hainan Island. In spring, *VR* = 2.58, which was greater than 1, indicating a positive association between the main fishes. The calculated statistic *W* was 128.79, which was not within the confidence interval of the chi-square distribution (34.76, 67.5), and the *VR* deviated significantly, indicating that there was a significant correlation among the nine main fish species in spring. In autumn, *VR* = 1.06, greater than 1, *W* = 53.11, which was within the confidence interval of the chi-square test (34.76, 67.5), indicating that the 17 main fishes in autumn showed a slight positive association (Table 4).

**Table 4.** Overall associations of major fishes in the inshore waters of Hainan Island.

| Season | $S_T^2$ | $\delta_T^2$ | VR | W | $\chi^2(\chi_{0.95}^2(50), \chi_{0.05}^2(50))$ | Inspection Result |
|---|---|---|---|---|---|---|
| Spring | 2.08 | 5.36 | 2.58 | 128.79 | (34.76, 67.5) | Significant positive association |
| Autumn | 3.84 | 4.08 | 1.06 | 53.11 | (34.76, 67.5) | Non-significant positive association |

*3.5. Interspecific Association Analysis*

The results of the chi-square test showed that among the 36 species pairs composed of 9 main fishes in spring (Figure 4a), 25 species pairs were not significantly associated ($\chi^2 < 3.841$), accounting for 69.44% of the total number of pairs; there were 11 species pairs with significant associations ($\chi^2 \geq 3.841$), among which 4 species pairs showed significant positive associations ($3.841 \leq \chi^2 \leq 6.635$, *ad > bc*), accounting for 11.11% of the total number of pairs; 1 species pair showed a significant negative association ($3.841 \leq \chi^2 \leq 6.635$, *ad < bc*), accounting for 2.78% of the total number of pairs; 6 species pairs showed extremely significant positive correlations ($\chi^2 > 6.635$, *ad > bc*), accounting for 16.67% of the total number of pairs; and there was no species pair with extremely significant negative associations ($\chi^2 > 6.635$, *ad < bc*). Among the 136 species pairs composed of 17 main fishes in autumn (Figure 4b), 91 species pairs were not significantly associated ($\chi^2 < 3.841$), accounting for 66.91% of the total number of pairs; there were 45 species pairs with significant associations ($\chi^2 \geq 3.841$), among which 5 species pairs showed significant positive associations ($3.841 \leq \chi^2 \leq 6.635$, *ad > bc*), accounting for 3.68% of the total number of pairs; 4 species pairs showed significant negative associations ($3.841 \leq \chi^2 \leq 6.635$, *ad < bc*), accounting for 2.94% of the total number of pairs; 15 species pairs showed extremely significant positive associations ($\chi^2 > 6.635$, *ad > bc*), accounting for 11.03% of the total number of pairs; and 21 species pairs showed extremely significant negative associations ($\chi^2 > 6.635$, *ad < bc*), accounting for 15.44% of the total number of pairs. In spring and autumn, a total of 56 species pairs had significant associations, of which 30 species pairs were positively associated, which were co-excellent species in the community, and had a high level of resource utilization and coincidence in the environment.

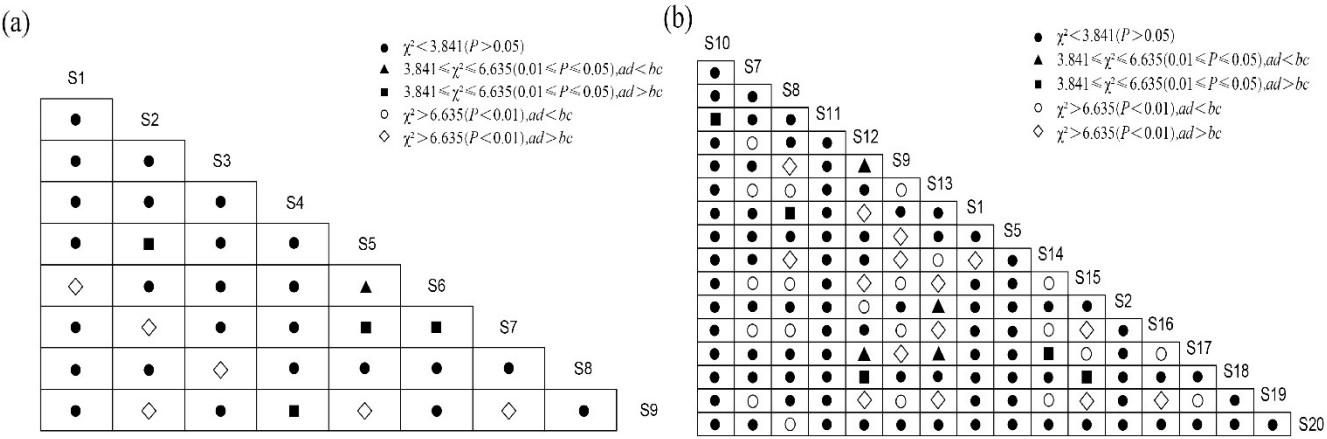

**Figure 4.** Semi-matrix of χ2 tests for major fishes in the inshore waters of Hainan Island. (**a**) For spring; (**b**) for autumn. The same below.

According to the association coefficient (*AC*), in spring (Figure 5a), 2 of the 36 species pairs had a strong positive association ($AC \geq 0.6$), accounting for 5.56% of the total number of pairs; 10 species pairs had an average positive association ($0.2 \leq AC < 0.6$), accounting for 27.78% of the total number of pairs; 19 species pairs tended to be independent from each other ($-0.2 \leq AC < 0.2$), accounting for 52.78% of the total number of pairs; 5 species pairs had an average negative association ($-0.6 \leq AC < -0.2$), accounting for 13.89% of the total number of pairs; and there was no species pair with strong negative association ($AC < -0.6$). In autumn (Figure 5b), 13 of the 136 species pairs had a strong positive association

($AC \geq 0.6$), accounting for 9.56% of the total number of pairs; 29 species pairs had an average positive association ($0.2 \leq AC < 0.6$), accounting for 21.32% of the total number of pairs; 52 species pairs tended to be independent of each other ($-0.2 \leq AC < 0.2$), accounting for 38.24% of the total number of pairs; 26 species pairs had an average negative association ($-0.6 \leq AC < -0.2$), accounting for 19.12% of the total number of pairs; and 16 species pairs had a strong negative association ($AC < -0.6$), accounting for 11.76% of the total number of pairs.

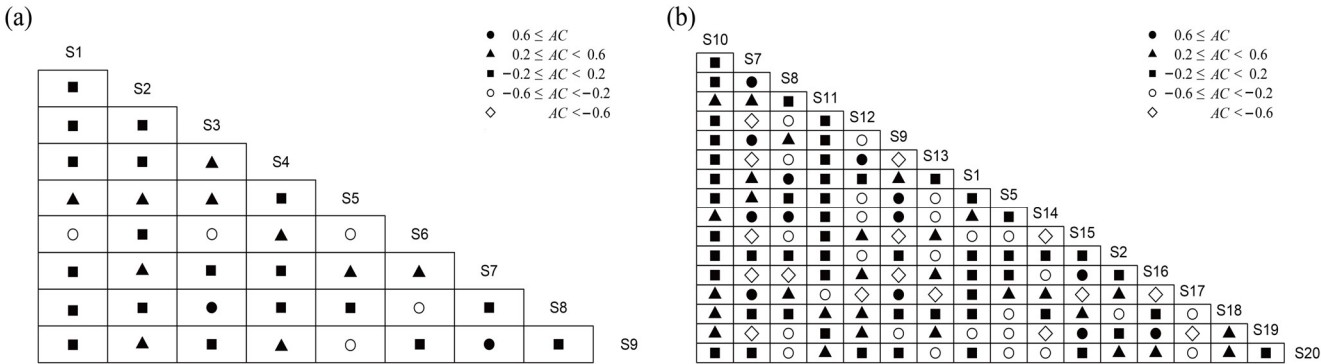

**Figure 5.** Semi-matrix of association coefficients (*AC*) for major fishes in the inshore waters of Hainan Island. (**a**) For spring; (**b**) for autumn. The same below.

According to the percentage of co-occurrence (*PC*), 5 of the 36 species pairs in spring (Figure 6a) had a close positive association ($PC \geq 0.6$), accounting for 13.89% of the total number of pairs; 21 species pairs had an average positive association ($0.4 \leq PC < 0.6$), accounting for 58.33% of the total number of pairs; 9 species pairs had a weak positive association ($0.2 \leq PC < 0.4$), accounting for 25% of the total number of pairs; only 1 species pair was not associated, accounting for 2.78%. In autumn (Figure 6b), 9 of the 136 species pairs were closely associated ($PC \geq 0.6$), accounting for 6.62% of the total number of pairs; 36 species pairs had an average positive association ($0.4 \leq PC < 0.6$), accounting for 26.47% of the total number of pairs; 49 species pairs had a weak positive association ($0.2 \leq PC < 0.4$), accounting for 36.03% of the total number of pairs; and 42 species were not associated, accounting for 30.88%.

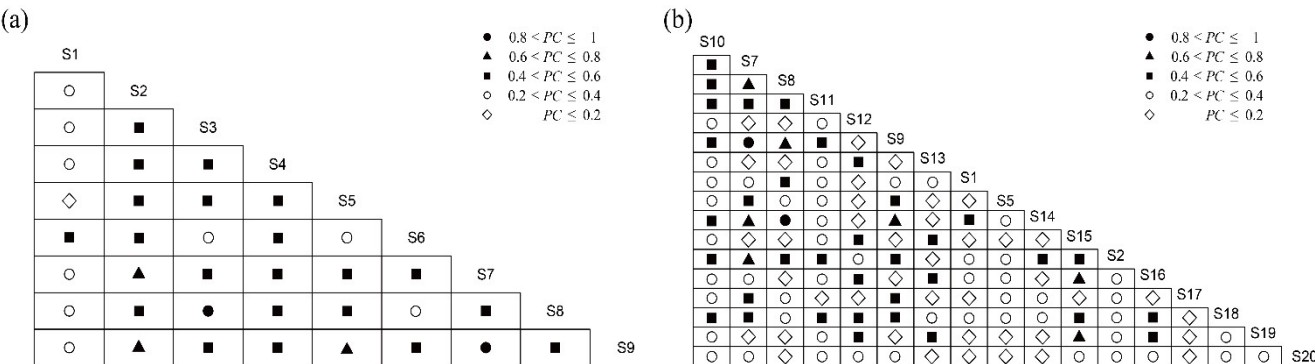

**Figure 6.** Semi-matrix of co-occurrence percentages (*PC*) of major fishes in the inshore waters of Hainan Island. (**a**) For spring; (**b**) for autumn. The same below.

### 3.6. RDA Graph Analysis

RDA was used to analyze the correlation between the main fishes and environmental factors pH, surface temperature (SST), bottom temperature (BST), surface salinity (SSS), bottom salinity (BSS), water depth, dissolved oxygen (DO), and chlorophyll a (Chl-a) that might influence niche differentiation. The results are shown in Figure 7.

In spring, surface temperature, water depth, and pH were significant environmental factors that affected the main fish species ($F = 4.4$, $p = 0.002$; $F = 3.4$, $p = 0.008$; $F = 2.5$, $p = 0.04$). The eigenvalues of Axis-1 and Axis-2 were 0.126 and 0.075, respectively, accounting for 20.14% of the species accumulation rate, and the correlation coefficient between species and environmental factors was 0.722 and 0.552, respectively, accounting for 26.2% of the total variation among the species data. Based on the Monte Carlo permutation test, there were significant differences between Axis-1 and Axis-2 ($F = 5.9$, $p = 0.036$; $F = 1.8$, $p = 0.018$), indicating that the ranking results passed the significance test and were reliable. Specifically, *T. hypargyreus* and *C. atridorsalis* were more sensitive to water depth and surface temperature, while *P. anomala* and *T. japonicus* were more sensitive to pH (Figure 7a).

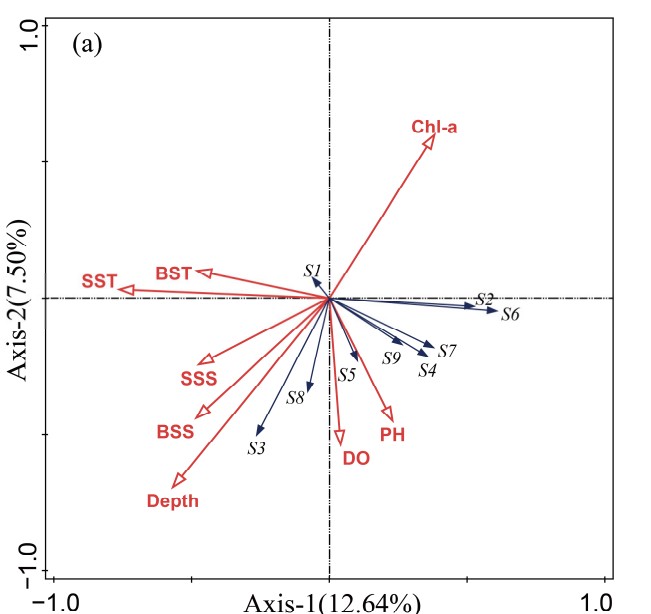 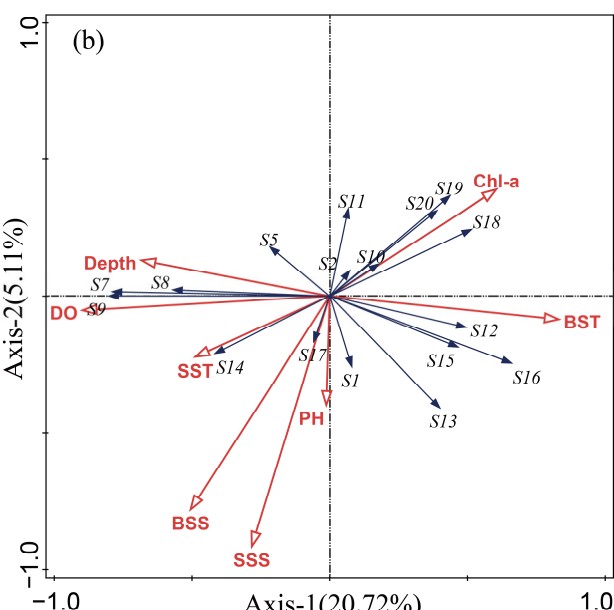

**Figure 7.** Redundancy analysis (RDA) of major fish and environmental factors in spring (**a**) and autumn (**b**). Bottom-layer salinity (BSS), surface-layer salinity (SSS), bottom-layer temperature (BST), surface-layer temperature (SST), water depth (Depth), pH, dissolved oxygen (DO), and chlorophyll-a concentration (Chl-a).

In autumn, the environmental factors that significantly affected the main fishes were dissolved oxygen, bottom temperature, surface salinity, and pH ($F = 9.8$, $p = 0.002$; $F = 2.8$, $p = 0.002$; $F = 3.0$, $p = 0.008$; $F = 1.9$, $p = 0.042$). The eigenvalues of Axis-1 and Axis-2 were 0.207 and 0.511, respectively, accounting for 25.83% of the species accumulation rate, and the correlation coefficient between species and environmental factors was 0.847 and 0.763, respectively, accounting for 35.4% of the total variation among the species data. The results of the Monte Carlo permutation test showed that the *p* values of Axis-1 and Axis-2 were both 0.002 ($F = 10.7$, $p = 0.002$; $F = 2.8$, $p = 0.002$). Specifically, the distribution of *S. tumbil* and *S. undosquamis* was greatly affected by dissolved oxygen; the distribution of *P. anea* and *Theraponidae* was greatly affected by the bottom temperature; and the distribution of *P. macrocephalus* and *Brachypleura novaezeelandiae* was greatly affected by surface salinity (Figure 7b).

## 4. Discussion

### 4.1. Niche Breadth

Niche breadth reflects a species' ability to adapt to habitats and use resources in a community [42,43]. An ecosystem is composed of a variety of species that occupy a variety of positions and play a variety of roles. While some species are able to adapt to a variety of environments and use a wide variety of resources, others are able to adapt to specific environments and use a small number of resources, which can be divided into broad ecosystems [43]. In this sense, species can be divided into broad niche species ($B_i \geq 2.571$) which have strong environmental adaptability and resource utilization, medium niche species ($2.027 \leq B_i < 2.571$) which have moderate ability, and narrow niche species ($0.297 < B_i < 2.027$) which have weak ability and can only use limited resources [40]. The niche breadth of the main fishes in the coastal waters of Hainan Island ranged from 0.297 to 3.293. The percentages of broad, medium, and narrow niche species were 20% and 40%, 10% and 30%, and 15% and 15%, respectively, among the main fishes in spring and autumn, indicating that broad niche species were the primary component of the main fishes in the area. In the study area, species with high abundance, wide spatial distribution, and high uniformity had a high degree of adaptability to changes in resources and environment. A species' niche breadth may vary according to its living habits and foraging behavior during different parts of the year [44]. *D. maruadsi*, for example, had the lowest niche breadth value in spring (0.621), making it a narrow niche species, but in autumn (2.773) it became a broad niche species, as it was observed at most stations in spring (the occurrence frequency was 62%). Nevertheless, its spatial distribution was extremely uneven (the abundance of *D. maruadsi* at station S3 accounted for 84.23% of the total). The autumn is also the main spawning season for *D. maruadsi*, as a large number of spawning populations gather and clusters appear [45]. Additionally, the high abundance of plankton in the surveyed sea area [46] and the summer moratorium provided sufficient food organisms and a good growth environment for juvenile fish, allowing them to spread widely while replenishing resources. *S. tumbil* and *S. undosquamis*, as important economic fish species in the coastal waters of Hainan Island, were found at most survey stations, which were broad niche species in both spring and autumn. According to the study, these species were widely distributed in the surveyed sea area, were highly reproductive, and had a high level of adaptability. They were capable of adapting quickly to changes in the resource environment, maintaining population stability, and distributing resources in a relatively balanced manner. According to Han et al. [47], niche breadth is not a sufficient indicator of species biomass. In this study, the dominant species were not those with higher niche breadth values. For example, *A. japonicum* was the first dominant species in spring but it was a narrow niche species in this season; The *IRI* of *C. atridorsalis* in spring was smaller than that of *T. japonicus*, but its niche breadth value was greater than the latter; the niche breadth value of *U. japonicus* in autumn was greater than that of *P. macrocephalus*, but its *IRI* was smaller than the latter. This suggested that there was no clear correlation between a species' *IRI* and niche breadth.

### 4.2. Niche Overlap

The niche overlap range of the main fishes in the coastal waters of Hainan Island was between 0 and 0.91. The species pairs with the highest niche overlap value in spring and autumn were both *S. tumbil* and *S. undosquamis*, both of which are warm-water demersal fish and belong to the same family and genus in taxonomy [48]. According to their feeding habits, they primarily consume *Sardinella spp.*, *Nemipterus spp.*, and *Decapterus spp.*, and the two also prey on each other's juveniles [49]. It would appear that when resources in its habitat were scarce, there was also an intense level of competition between species. Therefore, the shared habitat, similar bait composition, and the predation relationship between different species in the same ecosystem impacted the niche overlap between

them. In this regard, Li [50] reaches a similar conclusion. Moreover, Pratchett et al. [51] found that differences in feeding habits or habitats can reduce interspecific food competition between fish with similar ecological status. There was no niche overlap between *P. anea* and *R. asper* in autumn, which could be attributed to the differences in their feeding habits, adaptation to temperature and salinity, as well as their living habitats. Ge et al. [52] found a close relationship between the niche overlap value and species overlap on environmental sites. Based on the results of this study, the survey station overlap rate was 0% for both species, while it was 70% for *S. tumbil* and *S. undosquamis*. According to these results, the ecological niche overlap between the major fishes in Hainan Island's coastal waters was primarily determined by the similarity of habitat requirements, such as water layer, temperature, and salinity, and the similarity of predation requirements.

### 4.3. Overall Association and Interspecific Association

Interspecific association is a reflection of the stage and stability of community succession, while a static study of species association within a community can provide information on the dynamics of succession based on the relationship between species and their environment [53]. Therefore, the overall association is an important indicator of community structure and ecosystem stability. As indicated by the *VR* and value *W*, the overall association between the main fishes in the coastal waters of Hainan Island in spring was positive, indicating that species interacted closely and depended on each other, resulting in the community's stability and the ecosystem's health being maintained by their ecological niches and ecological functions. The positive association between the main fishes in autumn did not reach a significant level, suggesting a loose relationship between the main fishes. Biological and ecological characteristics of the species are mainly responsible for this looseness [54,55], and it may also be a result of the current stage of dynamic succession in the community.

Interspecific association can reveal species interrelationships and community dynamics and has been used by many scholars to study community succession and its dynamic processes [56,57]. Together, interspecific association and community stability reflect the stage and development of the community. As a result, when describing the characteristics and succession trends of the main fish communities in Hainan Island, they should be considered together. The results of the chi-square test showed that the insignificant species pairs in spring and autumn accounted for 69.44% and 66.91%, respectively, and the proportions of significant and extremely significantly associated species pairs were lower, which were 30.56% and 33.09%. In part, this may be due to the complex topography of the seabed near the coast of Hainan Island, which is characterized by shallow troughs, shoals, corrosion troughs, trenches, ridges, and hills [58–60]. Thus, communication between fish species is reduced, allowing each species to take up a more suitable living environment, resulting in a decrease in interspecific relationships. As well as internal disturbances, external disturbances, such as fishing and coastal engineering constructions, have a certain impact on interspecific associations [61]. Moreover, spring (27.78%) had a higher proportion of extremely significant and significantly positively associated species than autumn (14.71%), while autumn (18.38%) had a higher proportion of extremely significant and significantly negatively associated species than spring (2.78%). This is consistent with the conclusion that the main fish community is more stable in spring in the overall association analysis.

*AC* and *PC* are indicators used to measure the degree of association between species. The combination of *AC* and *PC* can be used to further evaluate the significance and strength of the interspecific association determined by the chi-square test. In addition, this method can be used to determine the validity and reliability of chi-square test results more accurately. Based on *AC* analysis, the proportion of positively associated species pairs was relatively high in spring and autumn, indicating that the community had a strong positive association, and species could coexist peacefully. The degree of positive association

between species is typically enhanced as a community matures [62]. Our *AC* analysis supports this conclusion. In this study, species pairs showing negative associations also accounted for a certain proportion, suggesting that the coastal marine ecosystem of Hainan Island was heterogeneous in some respects. Moreover, some species may choose to establish relationships with species with different functions, habitats, or living habits, increasing the number of negatively associated species pairs, which may result in relatively low relationships and connections between some species. *PC* analysis showed that there were five and nine species pairs with the strongest positive associations in spring and autumn, respectively, and the proportion of *S. tumbil*, *C. atridorsalis*, and *S. undosquamis* was relatively high. In part, this can be attributed to their relatively uniform spatial distribution, as well as their feeding habits [63]. *S. tumbil* and *S. undosquamis* are general-eating carnivorous fish, and other fish coexist as bait. They also have a higher competitive ability and survival ability than other fishes, thus the interspecific positive association is relatively stronger.

*4.4. The Relationship between Main Fish Niches and Environmental Factors*

There is a close relationship between the spatial distribution of fish communities and environmental factors, and the distribution of their ecological niches is influenced by environmental factors as well [64,65]. According to the RDA results, the main fishes in spring were significantly affected by environmental factors such as surface temperature (SST), water depth, and pH, while those in autumn were significantly affected by environmental factors such as dissolved oxygen (DO), bottom temperature (BST), surface salinity (SSS), and pH. The reason for this is that there are many rivers that flow into the sea along the coast of Hainan Island, bringing a large amount of nutrients and debris [66], which helps to improve the nutrient levels of the coastal waters. Furthermore, from July to September, the western boundary current of the South China Sea flows northward into the southern waters of Hainan Island and then northeastward, resulting in high temperature and high salinity [67]. As a result of this study, species with higher ecological niches, such as *S. tumbil*, *S. undosquamis*, and *B. novaezeelandiae*, were positively correlated with surface water temperature in autumn, supporting the point made above. In addition, Chl-a was another factor affecting the abundance of most fish species such as *P. edita*, *A.*, and *I. melastoma*. Chl-a concentration can be used as an indicator of phytoplankton productivity and biomass and is closely related to phytoplankton abundance [68]. The presence of zooplankton can also influence the growth and distribution of phytoplankton, thereby indirectly affecting the distribution of zooplankton [69]. Furthermore, the presence of zooplankton can provide adequate food for fish that consume zooplankton.

RDA can provide supplementary explanations for results that cannot be reasonably explained by ecological niches and their overlaps [70]. Generally, different living habitats may lead to low niche overlaps among fish species. In the present study, RDA results showed that the highest ecological niche species (*S. tumbil*) in autumn was mainly distributed in sea areas with a high DO concentration, while other species were distributed in sea areas with a lower DO concentration. For another example, though the niche breadths and ability to use resources in the habitat between *P. macrocephalus* and *B. novaezeelandiae* were quite different, the niche overlap value was high because they were both distributed in sea areas with high salinity. Consistent with our results, Dong et al. [71] reported that high similarity of environmental factors significantly enhanced the ecological overlaps of fish species in the southern coastal waters of Wenzhou, China.

## 5. Conclusions

This study analyzed the niche and interspecific relationships of fish community in the coastal waters of Hainan Island for the first time. Generally, broad niche species were the main components of this sea area, and the association among the main fishes was positive. In spring, the ranges of niche breadth and the niche overlap value were 0.621–3.039 and 0.001–0.91, respectively, while the corresponding values in autumn were 0.297–3.293 and 0–0.87, respectively. *AC* and *PC* tests revealed that the species were closely associated and generally exhibited a positive relationship. The results of RDA indicated that the community characteristics were significantly affected by environmental factors such as water temperature, depth, salinity, and pH value. To explore the mechanism of how the factors influence the structure and stability of the fish community in this sea area, further research should focus on monitoring their temporal niche over time and analyzing the stomach content using stable isotope methods.

**Supplementary Materials:** The following supporting information can be downloaded at https://www.mdpi.com/article/10.3390/fishes8100511/s1, Table S1: List of fish species in the coastal waters of Hainan Island in 2022.

**Author Contributions:** Conceptualization, Z.L.; original draft and data curation, Z.L.; funding acquisition, Y.L. and D.S.; validation, M.L. and C.C.; visualization, L.W. and B.S.; writing—review and editing, N.L., C.Y. and Y.Z. All authors have read and agreed to the published version of the manuscript.

**Funding:** This study was supported by the Hainan Provincial Natural Science Foundation of China (No. 320QN358), the Special Investigation on Basic Resources of Science and Technology (No. 2023FY100800), and the Central Public-interest Scientific Institution Basal Research Fund, CAFS (No. 2023TD93).

**Institutional Review Board Statement:** The animal study was reviewed and approved by the Committee of Laboratory Animal Welfare and Ethics of South China Sea Fisheries Research Institute on 11 September 2023. Approval code: nhdf2023-06.

**Informed Consent Statement:** Not applicable.

**Data Availability Statement:** The original contributions presented in the study are included in the article/Supplementary Material. Further inquiries can be directed to the corresponding authors.

**Conflicts of Interest:** The authors declare no conflict of interest.

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
