# Peer review of "Ecological Niche and Interspecific Association of the Main Fishes in the Coastal Waters of Hainan Island, China"

_fishes, doi:10.3390/fishes8100511_

Round 1

Reviewer 1 Report

General comments

The study uses bottom-trawl surveys conducted in the coastal waters of the Hainan Island in the spring and autumn of 2022 to explore the ecological niche and interspecific relationships among the main fish species. 

Several analytical methods, comprising association, classification and direct ordination were performed on the data to explore the organization of the fish community. Results revealed variation among the two seasons examined, including the influence of different explanatory/environmental variables in catch composition.

Overall, the authors have used appropriate sampling and statistical methods to assess the composition and organization of the fish community from the Coastal waters of the Hainan Island. Moreover, the importance of the study area and the considerable sampling effort make the results presented interesting to readers of Fishes.

Notwithstanding, I believe that the study could increase its global interest by making an effort to place the results in the context of studies addressing the organization of fish assemblages from other world coastal ecosystems. For example: i) are the levels of (positive and negative) association, or the levels of (high and low) overlap, common in other coastal fish communities? or ii) are the environmental variables related to the catches the same in other areas as the ones found in the study area?

I also suggest the authors to address cautiously in the discussion some of the results as evidence for factors structuring the fish community of the study area, although the authors rightly refer in the end that “To explore the mechanism of how the factors influence structure and stability of fish community in this sea area, further research should be focus on monitoring their temporal niche over time, and analyze the stomach content using stable isotope methods.”. For example, a high resource overlap could indicate competition only if resources are limited and if some population effects are evident (e.g. reduced growth, higher mortality rates, etc.). Otherwise, any suggestion of possible competitive interactions should be made with caution and be supported by results for the same taxa in other areas. Also, positive association should not be taken as indicative of higher levels of connectivity among fish species, as they can also indicate a common response to a particular resource.

There is a recently published study that uses the same data “Luo, Z.; Yang, C.; Wang, L.; Liu, Y.; Shan, B.; Liu, M.; Chen, C.; Guo, T.; Sun, D. Relationships between Fish Community Structure and Environmental Factors in the Nearshore Waters of Hainan Island, South China. Diversity 2023, 15, 901. https://doi.org/10.3390/d15080901” that should be referred and contextualized in the ms.

In conclusion, the study reported is interesting but should be revised to increase its overall interest to readers studying Coastal fish communities.

Please find below some minor comments that could help revise the ms.

Specific comments

Lines 68-70 “The fish community structure and biodiversity around Hainan Island have undergone major changes due to long-term heavy fishing [17], coastal engineering construction [18], and degrading water quality [19].” The references cited are not for the study area. Please rephrase the sentence.

Lines 78-79 “Additionally, redundancy analysis is employed to analyze the influence of environmental factors.” The influence of environmental factors on what?

Lines 82-83 “(2) exploring major fish species patterns of competitive relationships and resource allocation through (…)” patterns of “potential” competitive relationships.

Lines 86-90 “These research objectives contribute to enhancing our understanding of the ecological niche characteristics of major fish species, their species interactions, and the influence of environmental factors. This knowledge can positively impact the conservation and management of aquatic biological resources and the maintenance of ecosystem stability.” Please merge these two sentences in a single one.

Lines 93-94 “As China’s second largest offshore island (108°37′00″ – 111°03′00″E and 18°10′00″ - 20°10′00″N), Hainan Island is located” – As I understand this statement could be the object of dispute by some readers. Perhaps a different phrasing could be used.

Lines 376-378 “The niche breadth of the main fishes in the coastal waters of Hainan Island ranged from 0.297 to 3.293. The niche breadth of the main fish species in the coastal waters of Hainan Island ranged from 0.297 to 3.293.” The phrase is repeated, please delete one of the identical phrases.

Lines 409-411 “The term niche overlap refers to similarity or duplication of resource utilization between two or more species [46]. A niche overlap is also an important aspect of biodiversity, which is a reflection of the interaction and competition between different species and has a significant impact on the stability and function of an ecosystem.” This information should be placed either in the introduction or in the material and methods sections.

Please see the specific comments on the comments to authors.

Author Response

Responses to reviewer 1

The study uses bottom-trawl surveys conducted in the coastal waters of the Hainan Island in the spring and autumn of 2022 to explore the ecological niche and interspecific relationships among the main fish species.

Several analytical methods, comprising association, classification and direct ordination were performed on the data to explore the organization of the fish community. Results revealed variation among the two seasons examined, including the influence of different explanatory/environmental variables in catch composition.

Overall, the authors have used appropriate sampling and statistical methods to assess the composition and organization of the fish community from the Coastal waters of the Hainan Island. Moreover, the importance of the study area and the considerable sampling effort make the results presented interesting to readers of Fishes.

Notwithstanding, I believe that the study could increase its global interest by making an effort to place the results in the context of studies addressing the organization of fish assemblages from other world coastal ecosystems. For example: i) are the levels of (positive and negative) association, or the levels of (high and low) overlap, common in other coastal fish communities? or ii) are the environmental variables related to the catches the same in other areas as the ones found in the study area?

Response: Thank you for your valuable suggestion. The data presented in this study are derived from the most recent survey conducted in 2022, focusing on the nearshore waters of Hainan Island. To our knowledge, there is a scarcity of research exploring the ecological niches of fish species and their interspecific connectivity within this particular area. Moreover, upon conducting an extensive search for related literature, we observed that studies investigating the ecological niche and interspecific connectivity are predominantly concentrated in the field of botany, with fewer studies examining fish ecological niches within coastal ecosystems worldwide. To provide a more comprehensive understanding of the important ecological relationships, we will expand our data collection efforts by encompassing additional regions across the globe, alongside the coastal waters of Hainan Island in the further research.

Moving forward, we will enhance our survey efforts in the coastal waters of Hainan Island and extend the survey period (from 2022 onwards) to ensure the continued and comprehensive collection of data. Our aim is to gather more detailed survey information on the fish communities in this region, allowing us to conduct more extensive and in-depth investigations.

I also suggest the authors to address cautiously in the discussion some of the results as evidence for factors structuring the fish community of the study area, although the authors rightly refer in the end that “To explore the mechanism of how the factors influence structure and stability of fish community in this sea area, further research should be focus on monitoring their temporal niche over time, and analyze the stomach content using stable isotope methods.”. For example, a high resource overlap could indicate competition only if resources are limited and if some population effects are evident (e.g. reduced growth, higher mortality rates, etc.). Otherwise, any suggestion of possible competitive interactions should be made with caution and be supported by results for the same taxa in other areas. Also, positive association should not be taken as indicative of higher levels of connectivity among fish species, as they can also indicate a common response to a particular resource.

Response: Thank you very much for your review and suggestions. We very much agree with your views. However, when exploring situations where resources are limited and population effects are not significant (e.g., decreased growth rates, increased mortality, etc.), longer time spans and sufficient data, to support the study, are needed. The survey and data collection in the inshore waters of Hainan Island are still in the initial stage. In the future, we will strengthen the related work, collect more and more comprehensive first-hand data, and continue to analyze the interspecific relationships of the major fish species in the inshore waters of Hainan Island, as well as the specific factors and mechanisms affecting the interspecific relationships of the major fish species. We plan to apply methods of statistical analysis, such as regression modeling, in order to more accurately assess fish community dynamics and interactions. In addition, we plan to combine field surveys with simulation experiments to validate our findings. Through these efforts, we hope to provide a substantial scientific basis for ecological conservation and sustainable fisheries management in the nearshore waters of Hainan Island. Thank you again for your suggestions.

There is a recently published study that uses the same data “Luo, Z.; Yang, C.; Wang, L.; Liu, Y.; Shan, B.; Liu, M.; Chen, C.; Guo, T.; Sun, D. Relationships between Fish Community Structure and Environmental Factors in the Nearshore Waters of Hainan Island, South China. Diversity 2023, 15, 901. https://doi.org/10.3390/d15080901” that should be referred and contextualized in the ms.

Response: Thank you for your review and valuable comments on our paper. The data in our study aligns with the data sources mentioned in the aforementioned article. We omitted reference to a recently published study in our paper due to differences in research objectives, methods, and focus. The aforementioned article examines the relationship between fish community structure and environmental factors in the nearshore waters of Hainan Island. It employs the swept area method, the ecological character index method, and the ABC curve method to assess fish resource density and diversity. In the present study, we focus on assessing interspecific relationships among major fish species in the nearshore waters of Hainan Island, as well as examining the impacts of environmental factors on their ecological niches. The analysis employed several statistical methods, including relative importance index, ecological niche measure, cluster analysis, ratio of variance, χ2 test based on a 2×2 data matrix, coefficient of association, co-occurrence percentage, and redundancy analysis. Considering the relevance of the study to our research topic, based on your suggestion, we will add appropriate citations to the paper.

In conclusion, the study reported is interesting but should be revised to increase its overall interest to readers studying Coastal fish communities.

Please find below some minor comments that could help revise the ms.

Specific comments

  1. Lines 68-70 “The fish community structure and biodiversity around Hainan Island have undergone major changes due to long-term heavy fishing [17], coastal engineering construction [18], and degrading water quality [19].” The references cited are not for the study area. Please rephrase the sentence (L68 – L70).

Response: Thank you very much for your suggestions. Based on your feedback, we have made the necessary modifications to the respective section.

  1. Lines 78-79 “Additionally, redundancy analysis is employed to analyze the influence of environmental factors.” The influence of environmental factors on what? "

Response: Thank you for your valuable comments. We conducted a Redundancy Analysis (RDA) to evaluate the impact of environmental factors on major fish species. Meanwhile, we have incorporated the necessary revisions in the corresponding sections of the revised manuscript (L80).

  1. Lines 82-83 “(2) exploring major fish species patterns of competitive relationships and resource allocation through (…)” patterns of “potential” competitive relationships.

Response: Thank you very much for your advice. We have made change in the revised manuscript (L83).

  1. Lines 86-90 “These research objectives contribute to enhancing our understanding of the ecological niche characteristics of major fish species, their species interactions, and the influence of environmental factors. This knowledge can positively impact the conservation and management of aquatic biological resources and the maintenance of ecosystem stability.” Please merge these two sentences in a single one.

Response: Thank you very much for your guidance. Based on your suggestions, we have revised and adjusted the statements in the text (L87 – L90). Thank you again for your guidance and suggestions.

  1. Lines 93-94 “As China’s second largest offshore island (108°37′00″ – 111°03′00″E and 18°10′00″ - 20°10′00″N), Hainan Island is located” – As I understand this statement could be the object of dispute by some readers. Perhaps a different phrasing could be used.

Response: Thank you for your kind comments. We have made changes in the revised manuscript (L93).

  1. Lines 376-378 “The niche breadth of the main fishes in the coastal waters of Hainan Island ranged from 0.297 to 3.293. The niche breadth of the main fish species in the coastal waters of Hainan Island ranged from 0.297 to 3.293.” The phrase is repeated, please delete one of the identical phrases.

Response: Thank you for your valuable review comments. We sincerely apologize for the presence of a repetitive statement in the manuscript. We have thoroughly revised the content and sincerely appreciate your correction.

  1. Lines 409-411 “The term niche overlap refers to similarity or duplication of resource utilization between two or more species [46]. A niche overlap is also an important aspect of biodiversity, which is a reflection of the interaction and competition between different species and has a significant impact on the stability and function of an ecosystem.” This information should be placed either in the introduction or in the material and methods sections.

Response: Thank you for your valuable suggestions on our study. Based on your review comments, we have revised the paper accordingly. We have relocated this information to the Materials and Methods section to enhance the logical flow of the paper and align the relevant information with the research methodology (L139 - L142). Thank you again for your guidance.

Reviewer 2 Report

Thank you for the opportunity to review this paper. My line by line comments are in separate file.  

Author Response

Responses to reviewer 2

  1. Line 132. Please specify what do you mean by “in terms of its quality (%)”

Response: Thank you for your valuable comments. “in terms of its quality (%)” refers to the wet weight of a certain fish species and total catch, which we have explained in detail in the revised manuscript. Thank you again for your comments. (L131 – L134).

  1. Line 155. What do you mean by the overall variance of all stations? Which units you used? The same question for variance of all species.

Response: Thank you for your kind comments. The overall variance of all stations and all species refers to the frequency changes of station number and species number. We have made the corresponding changes in the revised manuscript (L160 – L164), and the two symbols are interpreted with reference to the paper: [Schluter, D. A Variance Test for Detecting Species Associations, with Some Example Applications. Ecology 1984, 65, 998-1005. ]. Since the calculation is based on frequency, the output does not include units.

  1. Line 158. “number of main fishes at station”. Do you mean species? Or individuals? Please specify.

Response: Thank you for your kind comments. In our study, "number of main fishes at station" mainly refers to species number of fish. We have specified it in the revised manuscript (L163).

  1. Linе 175-176. Please refer to the previous formula for a,b,с, d meaning.

Response: Thank you for your kind comments. Based on your comments, we have added the meaning of a, b, c and d in the revised manuscript (L188).
